# PairEdit: Learning Semantic Variations for Exemplar-based Image Editing

**Haoguang Lu**[1]    **Jiacheng Chen**[1]    **Zhenguo Yang**[2]    **Aurele Tohokantche Gnanha**[3]
**Fu Lee Wang**[4]    **Qing Li**[5]    **Xudong Mao**[1]*

[1]Sun Yat-sen University    [2]Guangdong University of Technology
[3]Huawei Noah's Ark Laboratory    [4]Hong Kong Metropolitan University
[5]The Hong Kong Polytechnic University

## Abstract

Recent advancements in text-guided image editing have achieved notable success by leveraging natural language prompts for fine-grained semantic control. However, certain editing semantics are challenging to specify precisely using textual descriptions alone. A practical alternative involves learning editing semantics from paired source-target examples. Existing exemplar-based editing methods still rely on text prompts describing the change within paired examples or learning implicit text-based editing instructions. In this paper, we introduce PairEdit, a novel visual editing method designed to effectively learn complex editing semantics from a limited number of image pairs or even a single image pair, without using any textual guidance. We propose a target noise prediction that explicitly models semantic variations within paired images through a guidance direction term. Moreover, we introduce a content-preserving noise schedule to facilitate more effective semantic learning. We also propose optimizing distinct LoRAs to disentangle the learning of semantic variations from content. Extensive qualitative and quantitative evaluations demonstrate that PairEdit successfully learns intricate semantics while significantly improving content consistency compared to baseline methods. Code is available at
`https://github.com/xudonmao/PairEdit`.

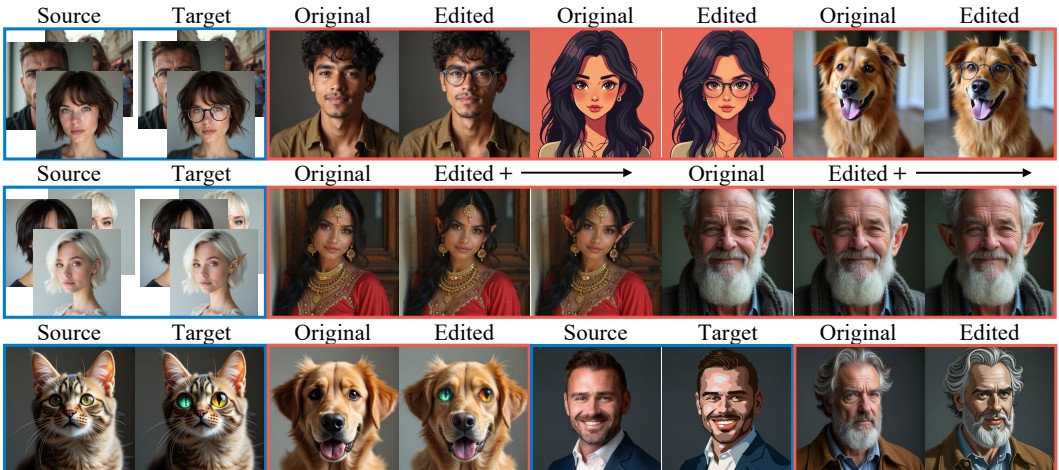

Figure 1: Editing results of PairEdit trained on three image pairs (1st-2nd rows) or a single image pair (3rd row). Our method effectively captures semantic variations between source and target images.

---

*Corresponding author (xudong.xdmao@gmail.com).

39th Conference on Neural Information Processing Systems (NeurIPS 2025).

# 1 Introduction

Recent advancements in diffusion models [23, 48] have significantly improved the quality and diversity of visual outputs, particularly in text-to-image synthesis tasks. The versatility of diffusion-based frameworks has further expanded their applicability beyond image generation into sophisticated image editing domains. Notably, text-guided editing has emerged as a powerful method, enabling fine-grained control over semantic attributes through natural language prompts [22, 42]. Additionally, diffusion models have been effectively employed in image-guided editing tasks [67, 10], facilitating the transformation of visual inputs guided by reference images, and in instructional editing tasks [5, 19], allowing intuitive edits through explicit instructions.

Among these, text-guided image editing has achieved remarkable success, enabling precise and flexible editing. Nevertheless, certain editing semantics are challenging to specify clearly through textual descriptions alone. A practical alternative involves learning semantics directly from paired images—consisting of before-and-after editing examples. However, existing exemplar-based editing methods typically rely on large language models or manual efforts to provide text prompts describing the change from source to target images [21, 9], or require encoding the change into the latent space of pre-trained instructional editing models [41, 56]. Notably, Concept Slider [18] introduces a loss function designed to train a single LoRA [24] with opposing scaling factors (positive and negative) to capture semantic variations by compelling predictions of identical noise. However, as illustrated in Figure 3, this method still struggles with learning complex semantics and maintaining content consistency between original and edited images.

In this paper, we introduce PairEdit, a novel visual editing method capable of effectively learning complex semantics from a small set of image pairs or even from a single image pair, without using any textual guidance. We explore optimizing LoRA to capture semantic variations between source and target images. To this end, we introduce a guidance-based noise prediction for LoRA optimization, explicitly modeling semantic variations by converting paired images into a guidance direction (i.e., $\epsilon_{\text{target}} - \epsilon_{\text{source}}$). Furthermore, we propose a content-preserving noise schedule designed to align the guidance scale with the LoRA scaling factor, enabling more effective semantic learning.

To disentangle semantic variation from content within paired images, we propose separating their learning processes by jointly optimizing two distinct LoRA modules: a content LoRA and a semantic LoRA. This optimization strategy encourages the content LoRA to reconstruct the source image while guiding the semantic LoRA to capture semantic variations from source to target images.

Our approach facilitates visual image editing based on a limited number of paired examples, effectively learning various semantics such as appearance change, age progression, and stylistic transformation. Moreover, our approach enables continuous control over the semantics by adjusting the scaling factor of the learned semantic LoRA. We demonstrate the effectiveness of our method through comprehensive qualitative and quantitative evaluations against several state-of-the-art methods. The results show that PairEdit achieves superior performance in terms of both identity preservation and semantic fidelity compared to existing baselines.

# 2 Related Work

**Text-to-Image Diffusion Models.** Diffusion models [54, 57, 23] have emerged as a dominant paradigm in text-to-image synthesis, which progressively refines Gaussian noise into high-quality images. In particular, latent diffusion models [48] employ U-Net architectures [49] to efficiently denoise in compressed latent spaces, setting the stage for notable improvements in resolution and scalability. Recent developments [14, 32] have initiated a shift from U-Net to vision transformer-based architectures, known as Diffusion Transformers (DiTs) [45]. These models utilize global attention mechanisms and advanced positional encodings to enhance model capacity and performance. These DiT-based diffusion models, such as Flux [32] and Stable Diffusion 3 [15], have consistently demonstrated state-of-the-art generation quality, with performance scaling predictably with model size. Moreover, flow-matching objectives [33, 34] have further enhanced the generation quality of these DiT-based models. Leveraging these advancements, Flux has achieved remarkable success in various applications such as image editing [13, 50, 61], personalized generation [16, 29, 66], and reference image generation [25, 37].

**Image Editing.** Generative adversarial networks [20, 38] have been extensively studied in the context of image editing by leveraging their expressive latent spaces [73, 52]. Recently, diffusion models, known for their superior capabilities in text-to-image generation, have attracted significant attention in image editing. Various input conditions have been investigated in diffusion-based image editing methods, as reviewed in [26]. Among these, text-guided image editing has achieved great success, offering an intuitive and flexible way for users to describe desired edits. This category includes approaches utilizing either descriptive texts for the edited image [22, 31, 30, 43, 60, 6, 44, 4] or explicit editing instructions [5, 19, 53, 72, 17, 27]. Additionally, some methods employ masks as input conditions to achieve precise control [69, 12, 63, 1, 2, 74], while others utilize reference images to guide the editing [70, 68, 58, 67, 10]. Another notable approach involves learning semantics directly from paired examples [3, 64, 18].

**Exemplar-based Image Editing.** Exemplar-based image editing has emerged as a powerful paradigm in image editing, effectively leveraging paired examples rather than relying on explicit textual instructions. MAE-VQGAN [3] first poses this problem as an image inpainting task, a framework that has since been adopted by several approaches [59, 64, 36, 21]. Alternative techniques utilize ControlNet-based architectures [70], as demonstrated by methods such as InstructGIE [40] and PromptDiffusion [65], which treat example images as spatial conditions. Other approaches build on the generalization capabilities of InstructPix2Pix [5] by inverting visual instructions into textual embeddings [41] or into LoRA weights [56]. Pair Customization [28] explicitly learns separate LoRAs for style and content within an image pair. Concept Slider [18] introduces a loss function that encourages a single LoRA with opposing scaling factors (positive and negative) to capture semantic variations by constraining them to predict the same noise. Despite these advancements, existing methods still face significant challenges in learning complex editing semantics from paired examples while maintaining content consistency between the original and edited images.

## 3 Method

### 3.1 Preliminaries

**Rectified-Flow Models.** Our approach is based on the Flux model, a type of rectified-flow model [33, 35] for text-to-image generation. Rectified-flow models define a transition from a Gaussian noise distribution $p_1$ to the real data distribution $p_0$. Given empirical observations from two distributions $x_0 \sim p_0$, $x_1 \sim p_1$, and $t \in [0, 1]$, the forward process of rectified-flow models is modeled as a continuous path:

$$x_t = (1 - t)x_0 + t\epsilon, \quad \epsilon \sim N(0, 1) \tag{1}$$

To reverse this process and recover data from noise, a velocity prediction network $v_\theta$ is trained to predict the velocity $v$ of the flow. This network can serve as a noise prediction network $\epsilon_\theta$ using the reparameterization technique introduced in [14].

**Classifier-Free Guidance.** Classifier-Free Guidance (CFG) is a technique introduced to improve the quality and controllability of samples generated by diffusion models without requiring an external classifier. CFG leverages the same prediction network $\epsilon_\theta$ in both conditional and unconditional modes. During sampling, predictions from the conditional model and the unconditional model are combined using a guidance scale $\gamma$:

$$\hat{\epsilon}_\theta = \epsilon_\theta(x_t, \varnothing) + \gamma \left( \epsilon_\theta(x_t, y) - \epsilon_\theta(x_t, \varnothing) \right). \tag{2}$$

The term $\epsilon_\theta(x_t, y) - \epsilon_\theta(x_t, \varnothing)$ is often referred to as the guidance direction. Our approach aims to construct a guidance direction corresponding to the target semantic variation observed within the paired images.

### 3.2 Learning Semantic Variations with PairEdit

Our goal is to learn semantic variations from a small set of image pairs. The key challenge lies in extracting accurate semantics that generalize well to editing new images. We introduce a novel LoRA-based method enabling precise and continuous image editing using only a few image pairs or even a single pair. Our method is based on three main ideas. First, we propose a guidance-based

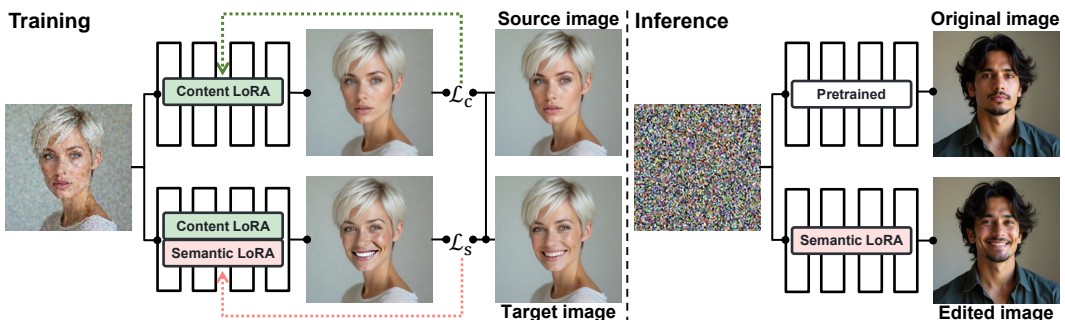

Figure 2: **Overview of PairEdit.** (Left) Given a pair of source and target images, we jointly train two LoRAs: a content LoRA, which reconstructs the source image using the standard diffusion loss (Eq. 3), and a semantic LoRA, which captures the semantic difference between the paired images using the proposed semantic loss (Eq. 10). (Right) During inference, when applying the learned semantic LoRA, the original image is edited towards the target semantic.

noise prediction that helps LoRA learn semantic variations from source to target images. Second, we introduce a content-preserving noise schedule for more effective semantic learning. Third, we propose separating semantic variation from content within image pairs by using two distinct LoRA adapters. An overview of the proposed PairEdit framework is depicted in Figure 2.

**Separating Semantic Variation and Content.** Our approach leverages the fact that paired images share the same content but differ only in target semantics. Inspired by recent studies in image stylization [28, 8], we jointly optimize two distinct LoRAs: a content LoRA, which reconstructs the source image, and a semantic LoRA, which captures semantic differences between source and target images. As illustrated in Figure 2, given the noised source image as input, the content LoRA aims to reconstruct the source image, while the semantic LoRA transforms the noised source image into the target image. Formally, we denote the content and semantic LoRA weights as $\theta_c$ and $\theta_s$, respectively. For the content LoRA, we employ a standard diffusion loss for reconstruction:

$$\mathcal{L}_{\text{content}} = \mathbb{E}_{x_0^A, \epsilon_0, t} \left[ \|\epsilon_0 - \epsilon_{\theta_c}(x_t^A, \varnothing)\|_2^2 \right], \tag{3}$$

where $x_0^A$ and $x_t^A$ denote the original and noised source images, respectively. For the semantic LoRA, however, we cannot simply rely on the reconstruction loss, as it involves denoising the noised source image toward the target image. Therefore, we explicitly model the semantic variation using a guidance direction term.

**Guidance-based Semantic Variation.** As illustrated at the bottom of Figure 2, we jointly leverage the content and semantic LoRAs to denoise the noised source image toward the target image. To achieve this, the predicted noise by these two LoRAs should incorporate the semantic variation from source to target images. Inspired by CFG (Eq. 2), we propose encoding the semantic variation into the CFG guidance direction. Thus, the target prediction noise $\epsilon^*$ for content and semantic LoRAs is defined as:

$$\epsilon^* = \epsilon_{\theta_c^*}(x_t^A, \varnothing) + \gamma(\epsilon_{\theta_{c,s}^*}(x_t^A, \varnothing) - \epsilon_{\theta_c^*}(x_t^A, \varnothing)), \tag{4}$$

where $\theta^*$ denotes the "ground truth" weights for content and semantic LoRAs, and $\gamma$ controls the strength of the guidance. In this equation, the first term $\epsilon_{\theta_c^*}$ corresponds to content reconstruction, while the guidance direction term $\epsilon_{\theta_{c,s}^*} - \epsilon_{\theta_c^*}$ corresponds to semantic variation. For simplicity, we denote $\epsilon_{\theta_c^*}(x_t^A, \varnothing)$ and $\epsilon_{\theta_{c,s}^*}(x_t^A, \varnothing)$ as $\epsilon_t^A$ and $\epsilon_t^B$, respectively. Note that the first term $\epsilon_{\theta_c^*}$ can be replaced with the true noise $\epsilon_0$ added to the source image. Thus, we reformulate Eq. 4 as:

$$\epsilon^* = \epsilon_0 + \frac{\gamma}{\Delta t}[(x_t^A - \Delta t \epsilon_t^A) - (x_t^A - \Delta t \epsilon_t^B)]. \tag{5}$$

Applying the denoising formula (i.e., $x_{t-\Delta t} = x_t - \Delta t \epsilon$), and considering that $\epsilon_t^B$ denoises the source image towards the target image, we derive:

$$\epsilon^* = \epsilon_0 + \frac{\gamma}{\Delta t}(x_{t-\Delta t}^A - x_{t-\Delta t}^B). \tag{6}$$

Utilizing Eq. 1 and applying identical noise to both source and target images, we obtain $x^A_{t-\Delta t} - x^B_{t-\Delta t} = (1 - t + \Delta t)(x^A_0 - x^B_0)$, yielding:

$$\epsilon^* = \epsilon_0 + \frac{\gamma}{\Delta t}(1 - t + \Delta t)(x^A_0 - x^B_0). \tag{7}$$

Here, the weight $\frac{\gamma}{\Delta t}(1 - t + \Delta t)$ is time-dependent. However, in practice, it is beneficial to establish a fixed weight aligned with a constant scaling factor of LoRA during optimization. To address this issue, we introduce a new noise schedule designed to make the weight of $x^A_0 - x^B_0$ time-independent.

**Content-Preserving Noise Schedule.** To achieve a time-independent weight for $x^A_0 - x^B_0$, we propose a new noise schedule defined as:

$$x_t = x_0 + t\beta\epsilon, \tag{8}$$

where $\beta$ controls the strength of the noise. Compared to the standard noise schedule (Eq. 1), our method preserves content information when $t = 1$; hence, we refer to it as the content-preserving noise schedule. Using this schedule, we derive $x^A_{t-\Delta t} - x^B_{t-\Delta t} = x^A_0 - x^B_0$ when applying identical noise to $x^A_0$ and $x^B_0$. Consequently, the target noise prediction becomes:

$$\epsilon^* = \beta\epsilon_0 + \eta(x^A_0 - x^B_0), \tag{9}$$

where $\eta = \frac{\gamma}{\Delta t}$.

As illustrated at the bottom of Figure 2, our semantic variation loss encourages the predicted noise $\epsilon_{\theta_{c,s}}$ by content and semantic LoRAs towards the target noise $\epsilon^*$, which is defined as:

$$\mathcal{L}_{\text{semantic}} = \mathbb{E}_{x^A_0, x^B_0, \epsilon_0, t}\left[\|\epsilon^* - \epsilon_{\theta_{c,s}}(x^A_t, \varnothing)\|^2_2\right]. \tag{10}$$

It is important to note that we optimize only the semantic LoRA weights with this loss, stopping gradient flow to the content LoRA weights. The benefits of our content-preserving noise schedule are two-fold. First, we set a fixed $\eta$ aligned with a constant scaling factor of the semantic LoRA, which stabilizes the training process. Second, for large $t$ values, our method preserves content information, resulting in meaningful semantic differences in $x^A_{t-\Delta t} - x^B_{t-\Delta t}$ (Eq. 6). In contrast, with the standard noise schedule, $x^A_{t-\Delta t} - x^B_{t-\Delta t}$ becomes meaningless as both $x^A_{t-\Delta t}$ and $x^B_{t-\Delta t}$ approach pure noise.

Although our noise schedule differs from the original one of the pretrained diffusion model, the pretrained model already has a general capability to handle and effectively denoise noisy inputs. Furthermore, LoRA can adapt the model's existing knowledge to this new noising approach. During inference, we follow the approach of [39, 18] by disabling the semantic LoRA for the initial $t$ steps to maintain content structure. Thus, the semantic LoRA does not need to learn denoising from purely noisy inputs.

Our full objective is:

$$\theta^*_s = \arg\min_{\theta_c, \theta_s} \mathcal{L}_{\text{content}} + \lambda\mathcal{L}_{\text{semantic}}, \tag{11}$$

where $\lambda$ controls the strength of the semantic loss.

## 4 Experiments

### 4.1 Implementation and Evaluation Setup

**Implementation Details.** Our implementation is based on the publicly available FLUX.1-dev[2], with both model weights and text encoders frozen. The rank of LoRA weights is set to 16. The parameter $\beta$ is set to 3 for global editing semantics (e.g., stylization) and 1 for local editing semantics (e.g., smile). For all experiments, $\eta$ and $\lambda$ are set to 4 and 1, respectively. We jointly train content and semantic LoRAs for 500 steps using a learning rate of $2 \times 10^{-3}$. The entire training process takes approximately 8 minutes on a single NVIDIA A100 80GB GPU. Following [39, 18], we set the LoRA scaling factor to 0 during the initial 14 steps to maintain the structure of the original image. Additional implementation details for our method and baseline methods are provided in Appendix A.

---

[2]`https://huggingface.co/black-forest-labs/FLUX.1-dev`

| Source | Target | Original | VISII | Transfer | Slider | Ours |
|--------|--------|----------|-------|----------|--------|------|

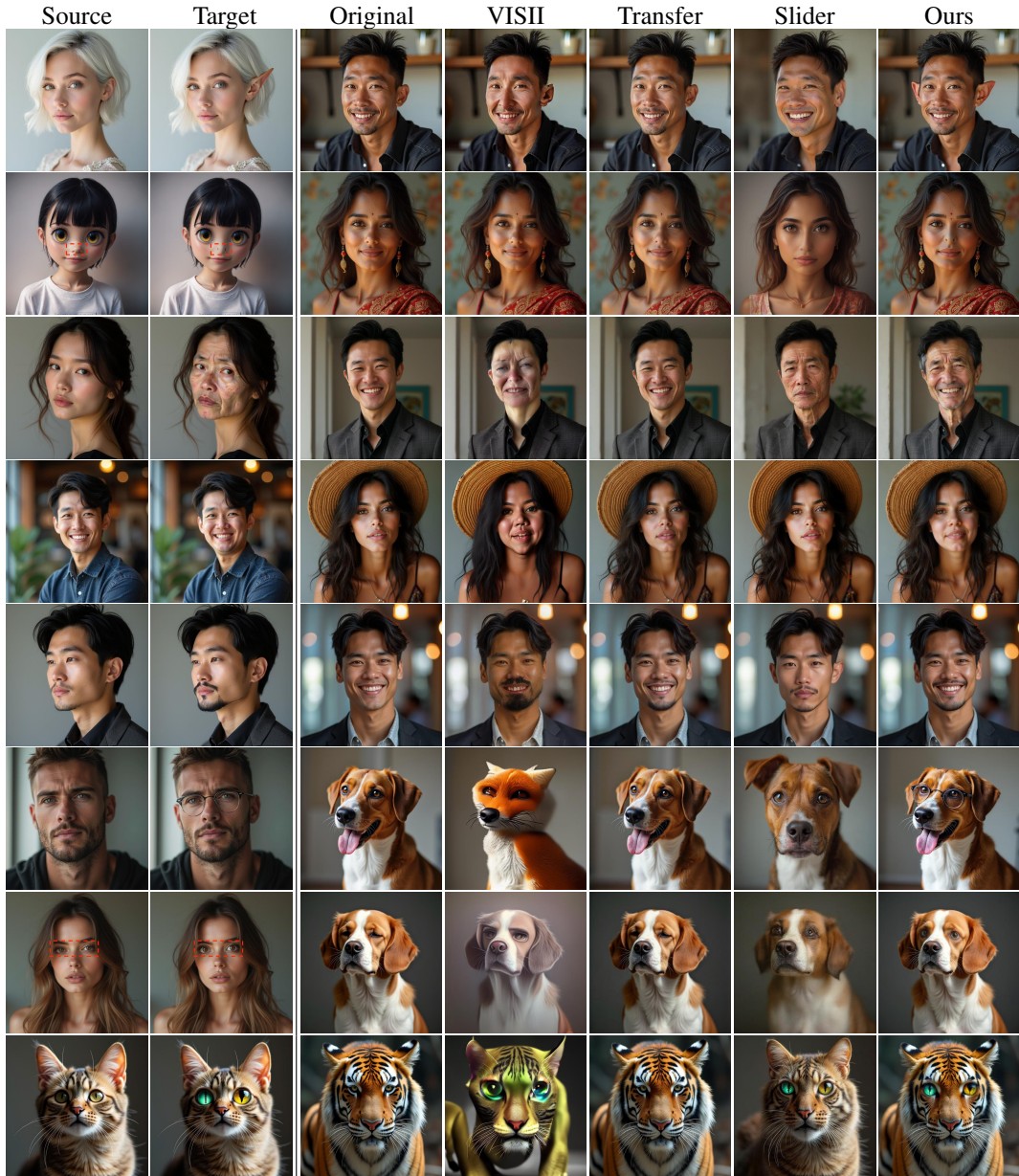

Figure 3: **Qualitative comparison.** We present exemplar-based image editing results of our method and three baseline methods, including VISII [41], Transfer [9], and Slider [18]. Our method demonstrates superior performance in accurately editing the original image while preserving its content.

**Datasets.** We create paired source and target images as follows: First, we apply existing image editing techniques, such as SDEdit [39], to translate source images into preliminary target images. Next, we transfer edited regions from the preliminary target images onto the corresponding regions of source images, generating the final target images. Additionally, some image pairs are collected from the web or sourced from [28]. For semantic learning, PairEdit is trained using either three image pairs (e.g., age, chubbiness, and elf ears) or a single image pair (e.g., stylization, lipstick, and dragon eyes).

**Evaluation Setup.** We compare our method with four exemplar-based editing methods: VISII [41], Edit Transfer [9], Pair Customization [28], and Visual Concept Slider [18], as well as two text-based editing methods that support continuous editing: SDEdit [39] and Textual Concept Slider [18]. For quantitative evaluation, we assess each method across four distinct semantics: age, smile, chubbiness,

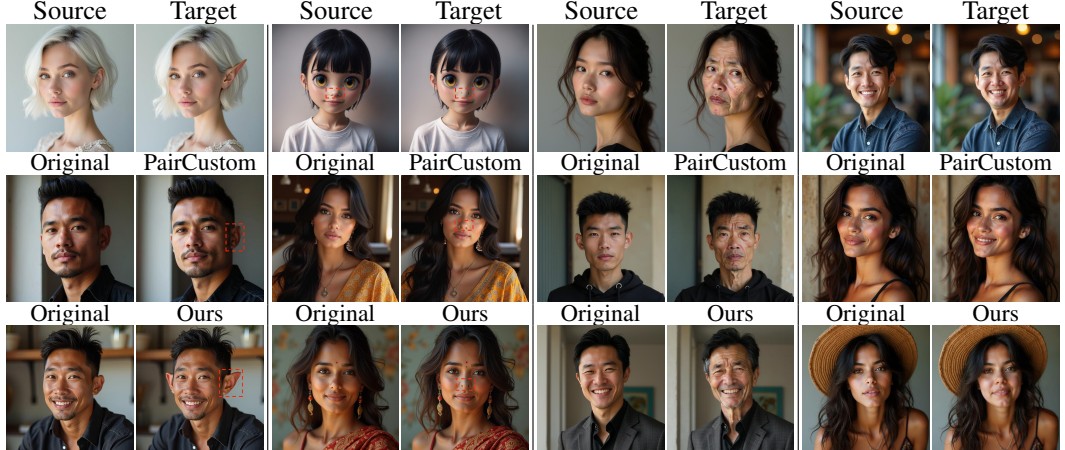

Figure 4: Qualitative comparison to Pair Customization [28]. As the official implementation of Pair Customization produces different outputs than the official FLUX.1-dev model for the same seed, we use different original images for comparison.

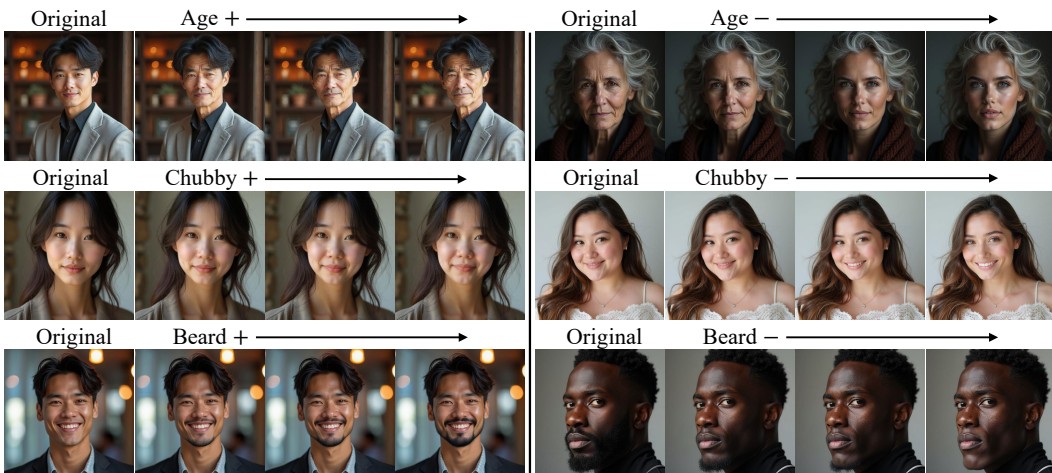

Figure 5: Examples of continuous editing by our method. By adjusting the scaling factor of the learned LoRA, our method enables a high-fidelity and fine-grained control over the semantic from exemplar images.

and glasses. For each semantic, we generate 500 pairs of original and edited images using the same random seed across all methods.

## 4.2 Results

**Qualitative Evaluation.** Figures 3 and 4 present visual comparisons of editing results between our method and the baselines. As the official implementation of Pair Customization [28] produces different outputs than the official FLUX.1-dev model for the same seed, we use different original images for comparison in Figure 4. We examine various editing tasks, including facial feature transformation, appearance alteration, and accessory addition. As shown, VISII [41] struggles to accurately capture semantic variations between source and target images, generating low-quality results. Edit Transfer [9] fails to capture semantic variations and produces images nearly identical to the original. Concept Slider [18] captures some semantic variations but consistently fails to preserve the identity of the original image. It also struggles with complex semantics (e.g., elf ears and chubbiness) and exhibits limited generalization capability (e.g., adding glasses to dogs). Pair Customization [28] fails to capture complex semantics and to preserve consistency with the original image. In contrast, PairEdit successfully performs all desired edits learned from paired examples.

| Real image | Reconst. | Age | Elf ear | Eye size | Chubby | Glasses |

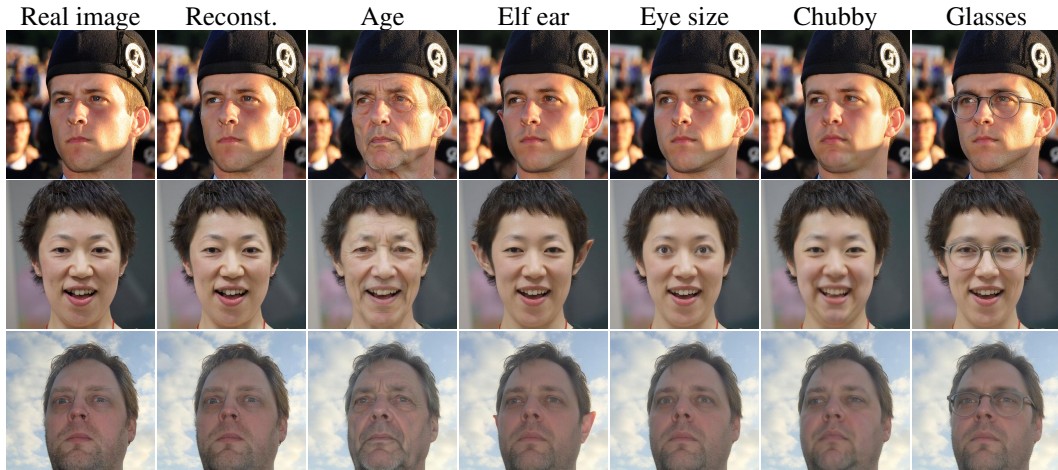

Figure 6: **Real image editing**. The reconstructed image is obtained by optimizing a LoRA over the real image. We apply the learned semantic LoRAs to the reconstructed image by merging the LoRAs during inference.

Table 1: **Quantitative comparison.** We evaluate each method by measuring identity preservation when performing similar editing magnitude. Identity preservation is measured using LPIPS distance, and editing magnitude is measured using the cosine similarity over CLIP embeddings.

| Semantics | SDEdit | | Textual Slider | | Visual Slider | | Ours | |
|---|---|---|---|---|---|---|---|---|
| | CLIP↑ | LPIPS↓ | CLIP↑ | LPIPS↓ | CLIP↑ | LPIPS↓ | CLIP↑ | LPIPS↓ |
| Age | 0.2285 | 0.1956 | 0.2266 | 0.1631 | 0.2257 | 0.1716 | **0.2382** | **0.1359** |
| Smile | 0.2533 | 0.1419 | 0.2556 | 0.1749 | 0.2724 | 0.1380 | **0.2896** | **0.1120** |
| Chubbiness | 0.2347 | 0.2173 | 0.2332 | 0.1423 | 0.2329 | 0.1747 | **0.2420** | **0.0815** |
| Glasses | 0.2419 | 0.1370 | 0.2427 | 0.1602 | 0.2421 | 0.1706 | **0.2886** | **0.0911** |

Moreover, our method achieves high-quality continuous editing by adjusting the scaling factor of the learned semantic LoRA, as illustrated in Figure 5. Additional qualitative evaluations are provided in Appendix B, and we also present a visual comparison of editing results using a single image pair in Appendix F.

**Quantitative Evaluation.** For quantitative assessment, we compare our method with three baselines that support continuous editing. We evaluate each method by measuring identity preservation while maintaining a similar editing magnitude. To ensure valid editing for the baselines, we employ a set of simple semantics for evaluation. Identity preservation is quantified using the LPIPS distance [71] between the original and edited images, while editing magnitude is measured via cosine similarity between CLIP embeddings [47] of the edited images and their corresponding textual editing descriptions. As demonstrated in Table 1, our method achieves significantly lower LPIPS distances compared to the baselines when applying comparable editing magnitudes. Additionally, we present DINO [7] results in Appendix C.

**User Study.** We also conducted a user study to evaluate our method. In each question, participants were shown a pair of source and target images, an original image, and two edited images: one produced by our method and the other by a baseline method. Participants were asked to select the image exhibiting superior editing quality while preserving the original identity. A total of 720 responses were collected from 24 participants, as detailed in Table 2. The results clearly indicate a strong preference for our method.

Table 2: **User Study.** Participants were asked to select the image exhibiting superior editing quality while preserving the identity.

| Baselines | Prefer Baseline | Prefer Ours |
|---|---|---|
| VISII [41] | 6.2% | **93.8%** |
| Analogist [21] | 1.3% | **98.7%** |
| Slider [18] | 3.8% | **96.2%** |

| Original | + Age | + Smile | + Lipstick | + Glasses | + Ear shape | + Eye color |

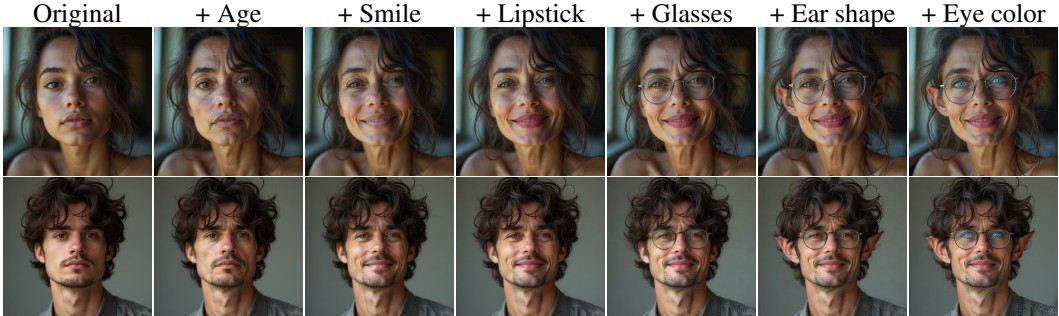

Figure 9: **Composing sequential edits.** Our method effectively composes different edits while preserving the original identity. Multiple semantic LoRAs are merged using the strategy illustrated in Eq. 12.

**Real Image Editing.** Editing real images typically involves finding an initial noise vector that reconstructs the input image using inversion techniques [55, 51]. However, we observe that directly applying existing inversion methods designed for Flux [51] with the learned semantic LoRA yields poor editing quality. This issue arises because these inversion methods fail to accurately map the input image back into Flux's original latent space, a limitation also highlighted in [13]. Since inversion methods are not the primary focus of this paper, we adopt a simple reconstruction strategy by optimizing a LoRA over the input image. To apply learned edits to reconstructed images, we merge the two LoRAs during inference as follows:

$$\epsilon_{\theta_r}(x_t, \varnothing) + \gamma_{\text{real}}(\epsilon_{\theta_s}(x_t, \varnothing) - \epsilon_{\theta_r}(x_t, \varnothing)), \tag{12}$$

where $\theta_r$ and $\theta_s$ denote the reconstruction and semantic LoRA weights, respectively, and $\gamma_{\text{real}}$ is set to 0.75. We empirically find that this merging strategy improves identity preservation compared to linear combination of LoRA weights, as illustrated in Appendix E. As shown in Figure 6, our approach achieves high-quality editing while effectively preserving the identity of real images.

**Composing Sequential Edits.** Our method supports combining multiple edits through the merging of several learned semantic LoRAs. We employ the same merging strategy during inference as in real image editing, resulting in better editing quality compared to a linear combination of LoRA weights. As illustrated in Figure 9, our method effectively composes multiple edits while preserving individual identities.

**Weakly Aligned Image Pairs.** For weakly aligned image pairs, our model can still learn semantic variations. However, it may capture additional unintended semantics, as it cannot explicitly identify which semantics are targeted when only 1–3 pairs are used for training. High-quality image pairs lead to improved generation quality, particularly in terms of identity preservation.

| Source | Target | Original | Edited |

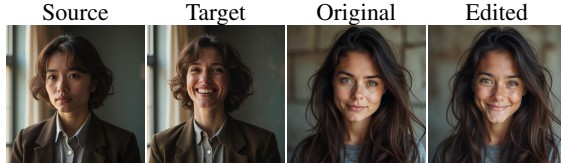

Figure 7: Weakly aligned pairs.

Figure 7 shows an example trained on weakly aligned image pairs. The edited image exhibits inconsistencies in certain aspects, such as background and hair appearance.

**Knowledge Transfer Across Different Edits.** In Figure 8, we demonstrate the transfer of a learned LoRA (adding glasses) to a different edit (increasing eye size). The model successfully learns the new edit while significantly reducing optimization steps from 500 to 50. This demonstrates that knowledge encoded in learned LoRAs can be effectively reused to accelerate the learning of related edits.

| Original | 50 steps | 500 steps | 50 steps Transferred |

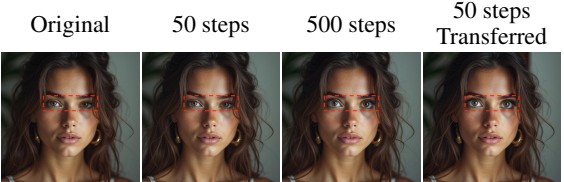

Figure 8: Learning "increasing eye size" by transferring from a learned "adding glasses" LoRA.

| Source | Target | Original | Variant A | Variant B | Variant C | Ours |
|--------|--------|----------|-----------|-----------|-----------|------|

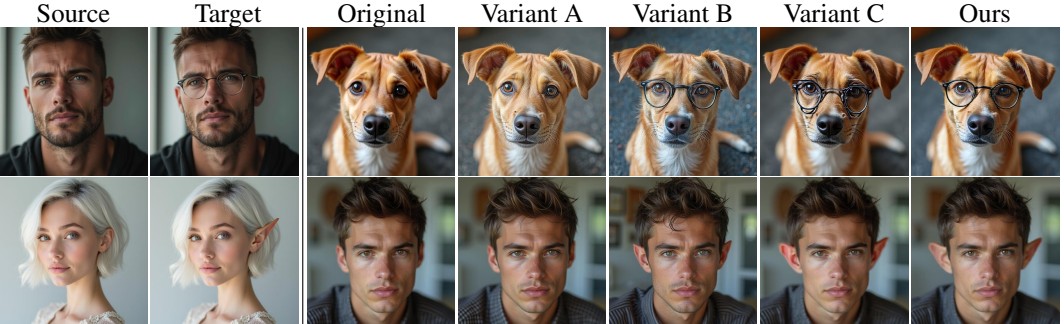

Figure 10: **Ablation study.** We evaluate three variants of our model: (A) replacing the semantic loss with the visual concept loss proposed in [18], (B) removing the content LoRA, and (C) replacing the content-preserving noise schedule with a standard noise schedule.

## 4.3   Ablation Study

In this section, we perform an ablation study to assess the effectiveness of individual components within our framework. Specifically, we evaluate three variants: (1) replacing the semantic loss with the visual concept loss proposed in [18], (2) removing the content LoRA, and (3) replacing the content-preserving noise schedule with a standard noise schedule. Figure 10 presents a visual comparison of editing results generated by each variant. The results demonstrate that all proposed components are crucial for achieving identity preservation and semantic fidelity. Omitting our semantic loss significantly reduces the model's ability to capture complex editing semantics. Removing the content LoRA leads to inconsistent results, such as unintended fur color changes in the first row and hairstyle alterations in the second row. Employing a standard noise schedule negatively affects the generalization capability of the semantic LoRA, causing blurred glasses in the first row and inconsistent ear coloration in the second row. Additional ablation study results are provided in Appendix G.

## 5   Conclusions and Limitations

In this paper, we introduced PairEdit, a novel visual editing framework designed to effectively capture complex semantic variations from limited paired-image examples. Utilizing a guidance-based target denoising prediction term, our method explicitly transforms semantic differences between source and target images into a guidance direction. By separately optimizing two dedicated LoRAs for semantic variation and content reconstruction, PairEdit effectively disentangles semantic attributes from content information. However, one limitation of PairEdit is its reliance on paired images, which is not directly applicable to unpaired datasets. In future work, we aim to explore methods capable of extracting editing semantics from unpaired image sets, further enhancing the flexibility and practical applicability of our approach.

## Acknowledgments

This work is supported by National Natural Science Foundation of China (No. 62176223 and No. 62302535), Guangdong Basic and Applied Basic Research Foundation (No. 2023A1515012897), and Zhuhai Basic and Applied Basic Research Foundation (No. 2320004002745).

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

# A    Implementation Details.

Our method leverages FLUX.1-dev, with both model weights and text encoders fixed. We employ the Adam optimizer to tune the LoRA weights, setting the rank to 16. The content and semantic LoRAs are jointly trained for 500 steps using a learning rate of $2 \times 10^{-3}$. For all experiments, we perform image generation with 28 inference steps. To preserve the structure of the original image, we follow the approach described in [39, 18], setting the LoRA scaling factor to 0 for the initial 14 steps. For Textual Concept Slider [18], we utilize their official Flux implementation. For Visual Concept Slider [18], due to the unavailability of the official Flux implementation, we implement the Flux-based model following their SDXL implementation. For other baseline methods, including VISII [41], Analogist [21], and Edit Transfer [9], we utilize their official implementations and follow the hyperparameters described in their papers. For SDEdit [39], we use the diffusers Flux implementation. When using GPT-4o, the editing prompt is: "The first and second images represent a 'before and after' editing pair. Please analyze the changes made between them and apply the same edit to the third image."

# B    Additional Qualitative Results

In Figure 11, we present additional qualitative comparisons against three baseline methods: Edit Transfer [9], GPT-4o, and Visual Concept Slider [18]. Our method demonstrates superior performance in terms of identity preservation and semantic fidelity compared to the baselines. Edit Transfer struggles to accurately capture the semantic variations between source and target images. GPT-4o shows poor identity preservation, and Visual Concept Slider also fails to preserve the original identity while struggling with complex semantic edits.

# C    Additional Quantitative Results

In addition to the CLIP and LPIPS metrics, we present results using the DINO metric in Table 3. Our method achieves superior performance compared to the baselines in terms of DINO score.

# D    Additional Real Image Editing Results

In Figure 12, we provide additional examples of real image editing. Reconstructed images are obtained by optimizing LoRAs directly over real images. We apply the learned semantic LoRAs to these reconstructed images using guidance-based LoRA fusion as described in Eq. 12. The results demonstrate the effectiveness of our approach in editing real images across various semantic attributes.

# E    Comparison of LoRA Fusion Methods

In Figure 13, we compare two LoRA fusion methods: (1) linear combination of LoRA weights and (2) guidance-based LoRA fusion (Eq. 12). The linear combination approach tends to produce blurry outputs for certain semantics, whereas the guidance-based LoRA fusion provides better identity preservation and image quality.

# F    Learning with a Single Image Pair

In this section, we evaluate PairEdit when trained using only a single image pair. As shown in Figure 14, our method outperforms baseline methods in both identity preservation and semantic fidelity. However, we observe that providing multiple image pairs further helps the model learn complex semantics and enhances its generalization capability (e.g., adding glasses to dogs).

# G    Additional Ablation Study

As discussed in Section 4.3, we evaluate three variants of our model: (1) replacing the semantic loss with the visual concept loss from [18], (2) removing the content LoRA, and (3) substituting the

content-preserving noise schedule with a standard noise schedule. Additional results of the ablation study are presented in Figure 15.

## H    Training Data

For the paired training samples, the target image is created by coarsely copying the target region into the source image. Even though the pasted target images often have obvious artifacts around the boundaries, we find that our model is able to effectively learn the semantic edits while ignoring these inconsistencies. In Figure 16, we present examples of our training data. Our model is trained using either three image pairs (e.g., elf ears, glasses, and chubbiness) or a single image pair (e.g., stylization, dragon eyes, and lipstick).

## I    Implementation with SDXL

In this section, we evaluate the performance of our model with SDXL [46] as the backbone. As shown in Figure 17, the model remains capable of learning meaningful editing semantics. However, its performance is not as strong as with FLUX in terms of identity preservation and editing quality. This performance gap stems from differences in the mathematical modeling of the backbone models—our approach is theoretically grounded in the noise formulation (Eq. 1) used in FLUX from a flow-matching perspective, whereas SDXL employs a DDPM [23] noise schedule.

## J    Failure Cases

When the source and target images exhibit significant structural differences, our method may struggle to capture the semantic variations. For example, as illustrated in Figure 18, when a person's pose changes from arms hanging naturally to arms crossed over the chest, our model cannot learn this transformation. Such transformations exceed the intended scope of our method.

## K    User Study

As described in Section 4.2, we conducted a user study to evaluate our method against the baselines. Figure 19 shows an example question from the user study. Given a pair of source and target images, an original image, and two edited images: one produced by our method and the other by a baseline method. Participants were asked to select the image exhibiting superior editing quality while preserving the original identity. The results are presented in Table 2.

## L    Societal Impact

Similar to existing image editing techniques, our approach enables users to effectively edit images by optimizing LoRA weights of large-scale pre-trained diffusion models. By allowing individuals to manipulate images using their own data, this method supports a wide range of applications, such as novel content generation and artistic creation. Despite these positive outcomes, the use of generative models also introduces risks, including the creation of misleading or false information. To address these concerns, it is essential to advance reliable detection methods for distinguishing real images from synthetic ones [62, 11].

## M    Licenses for Pre-trained Models and Datasets

Our implementation is based on the publicly available FLUX.1-dev, which is licensed under the FLUX.1-dev Non-Commercial License. Most of the images used for evaluation are created using FLUX.1-dev and SDEdit [39]. Some image pairs are collected from the web or sourced from [28]. The license information for these images is not available online.

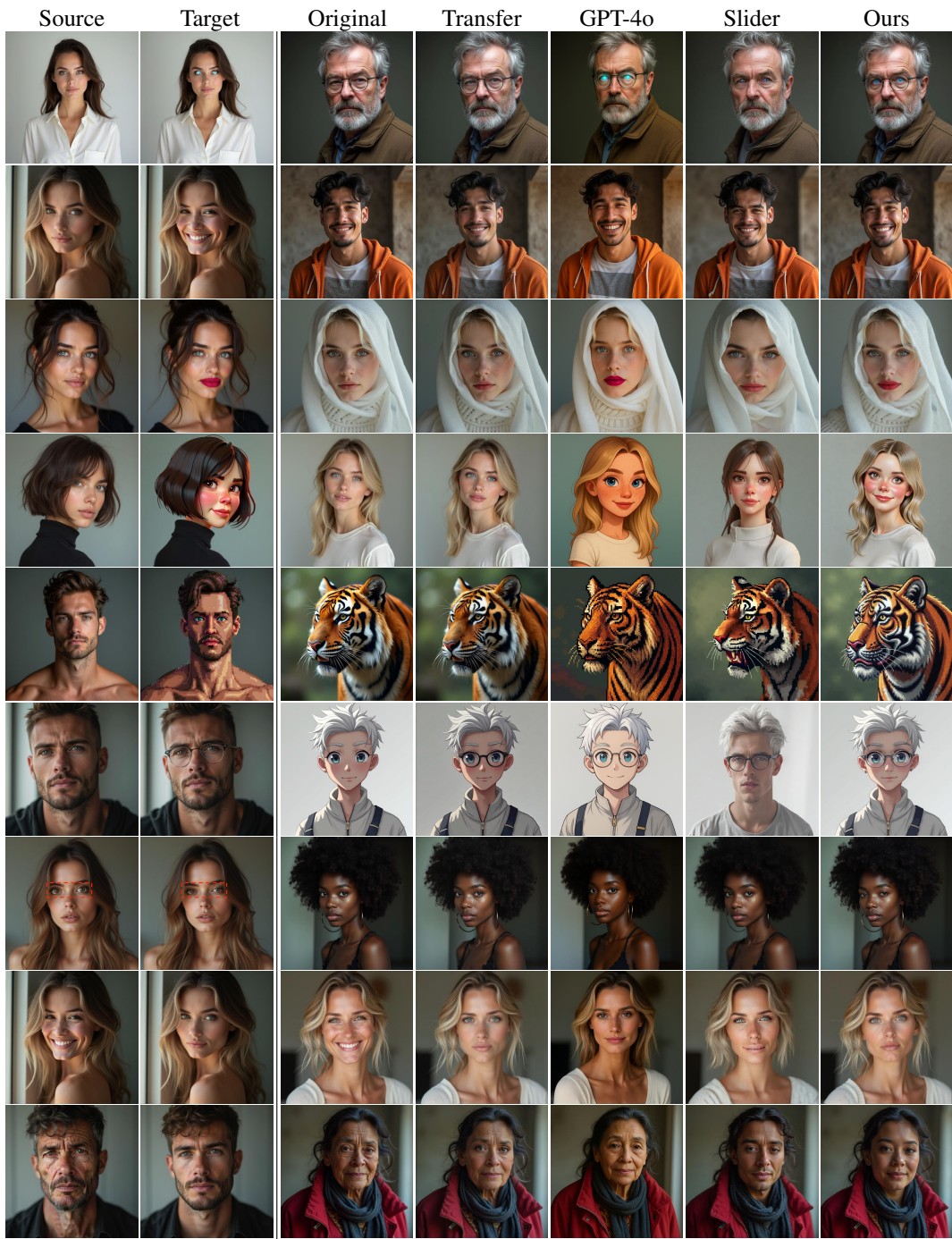

Figure 11: **Additional qualitative comparison.** We present exemplar-based image editing results from our method and three baseline methods: Edit Transfer [9], GPT-4o, and Slider [18]. Our method demonstrates superior performance in accurately editing the original image while preserving its content.

| Real image | Reconst. | Smile | Eye size | Chubby | Elf ear | Lipstick |
|---|---|---|---|---|---|---|

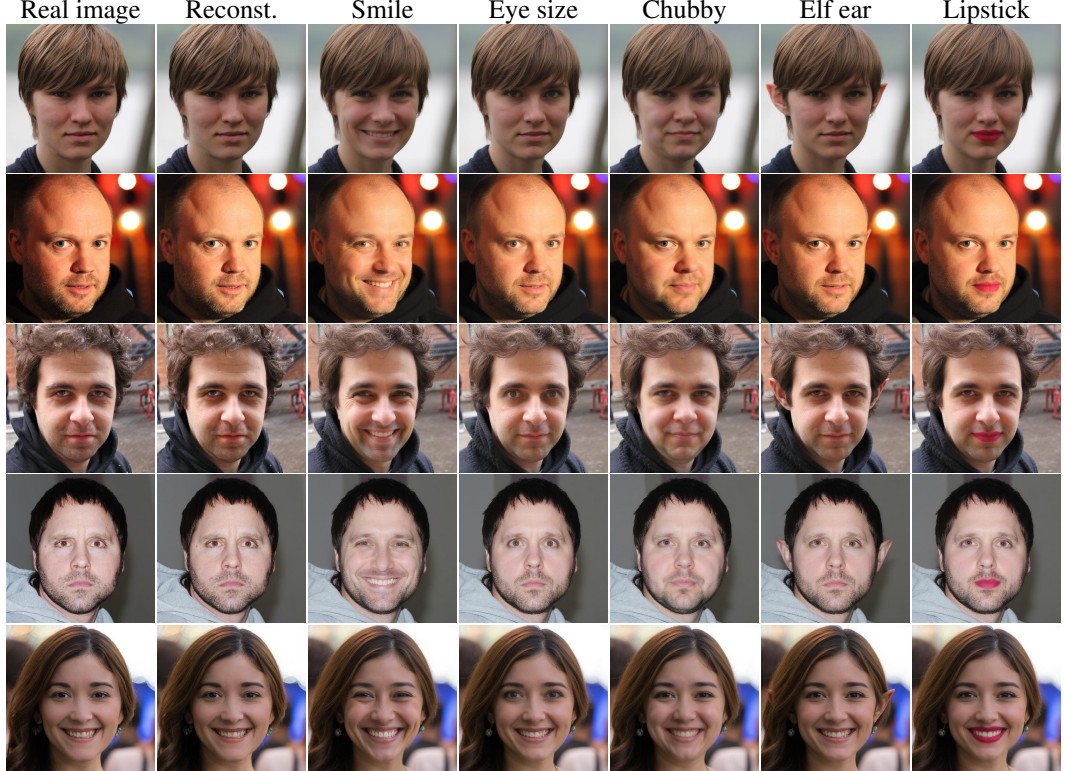

Figure 12: **Additional real image editing results.** The reconstructed image is obtained by optimizing a LoRA on the real image. We apply the learned semantic LoRAs to the reconstructed image by merging the LoRAs during inference.

| Real image | Linear Comb. | Ours | Real image | Linear Comb. | Ours |
|---|---|---|---|---|---|

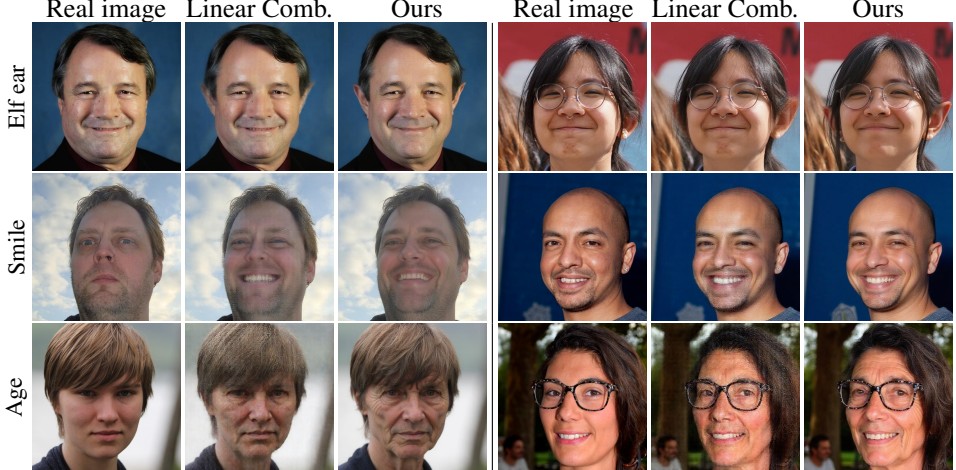

Figure 13: Comparison of two LoRA fusion methods: (1) linear combination of LoRA weights and (2) guidance-based LoRA fusion (Eq. 12). Guidance-based LoRA fusion achieves better identity preservation, whereas linear combination of LoRA weights tends to generate blurry images for certain semantics.

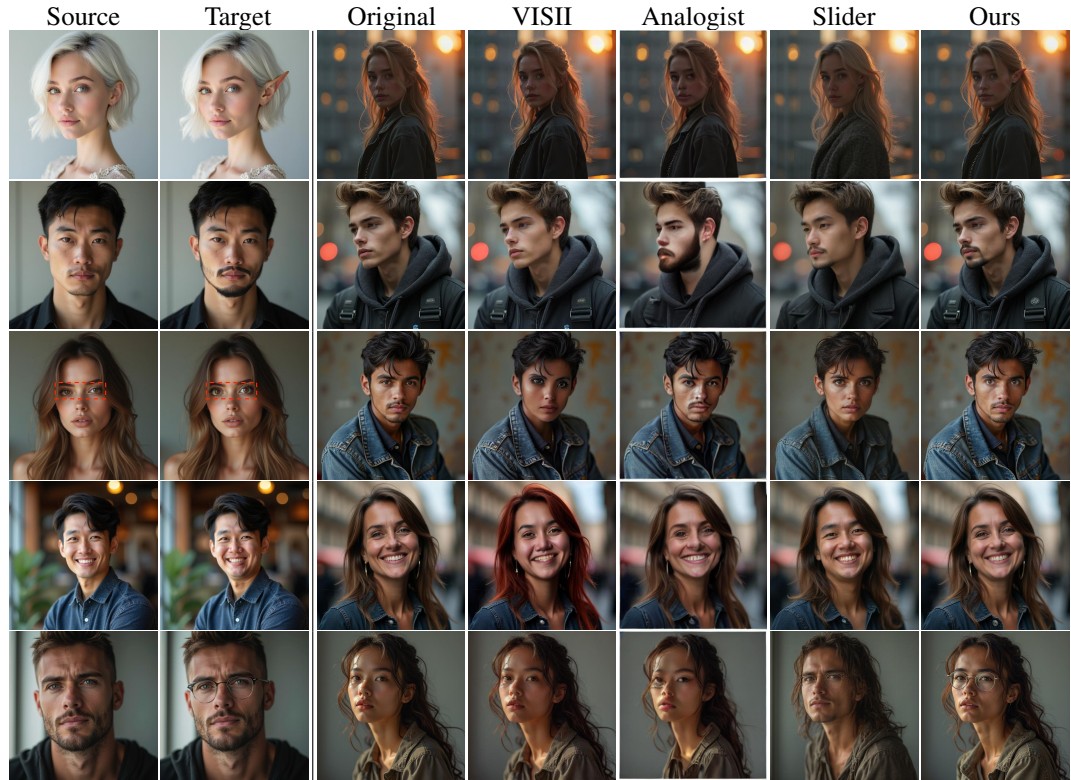

Figure 14: Comparison of PairEdit with three baseline methods under a single-image-pair training setting. Our method demonstrates superior performance in both identity preservation and semantic fidelity.

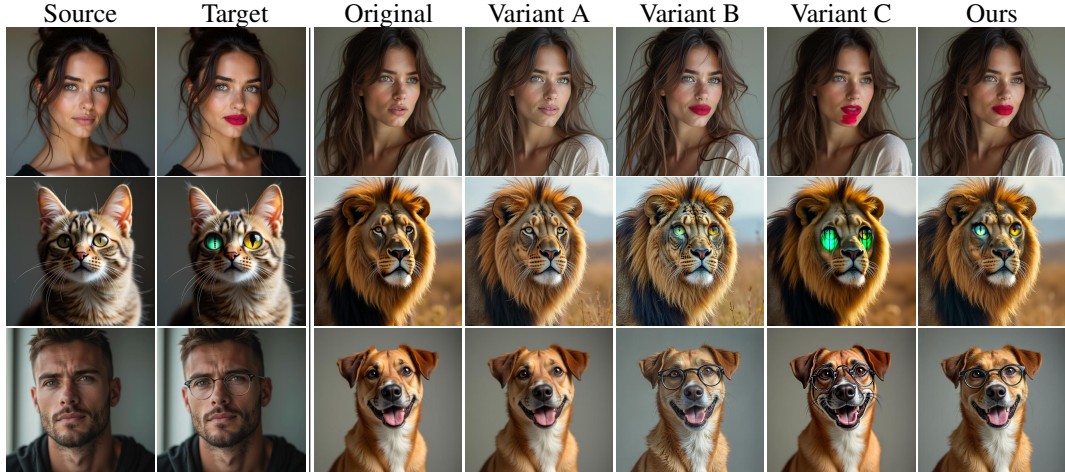

Figure 15: **Additional ablation study results.** We evaluate three variants of our model: (A) replacing the semantic loss with the visual concept loss proposed in [18], (B) removing the content LoRA, and (C) replacing the content-preserving noise schedule with a standard noise schedule.

Table 3: Quantitative comparison using the DINO metric [7].

| DINO↑ | SDEdit | Textual Slider | Visual Slider | Ours |
|---|---|---|---|---|
| Chubbiness | 0.8420 | 0.9152 | 0.8882 | **0.9588** |
| Glasses | 0.8408 | 0.8951 | 0.8810 | **0.9065** |
| Smile | 0.9086 | 0.8853 | 0.9143 | **0.9150** |
| Age | 0.8597 | 0.8875 | 0.8564 | **0.8885** |

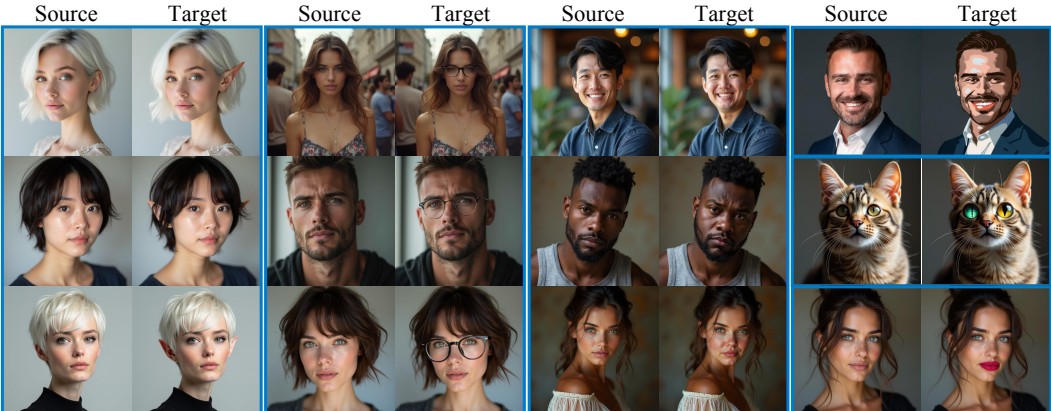

Figure 16: **Examples of training data**. Our model is trained using either three image pairs (e.g., elf ears, glasses, and chubbiness) or a single image pair (e.g., stylization, dragon eyes, and lipstick).

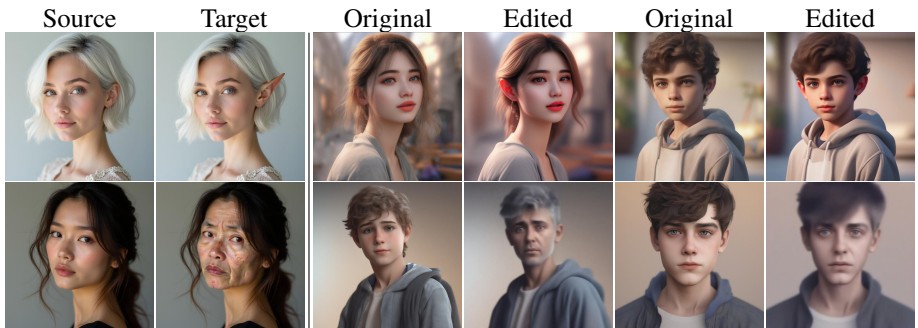

Figure 17: Results of our model with SDXL [46] as the backbone.

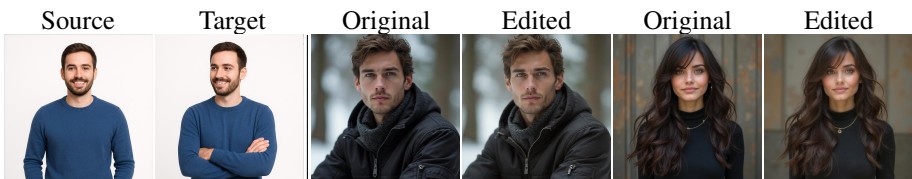

Figure 18: **Failure examples**. When the source and target images exhibit significant structural differences, our method may struggle to capture the semantic variations.

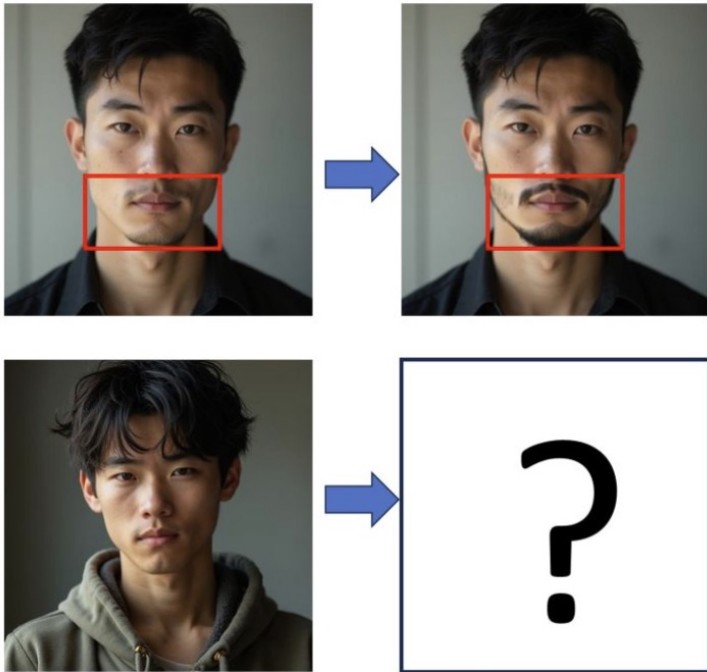

First Row of Images: Observe the editing effects from the left image to the right image (highlighted in the red box).

Our goal is to apply the editing effect from the first row to the image in the second row.

Please choose the image below that achieves superior editing quality while preserving the original identity.

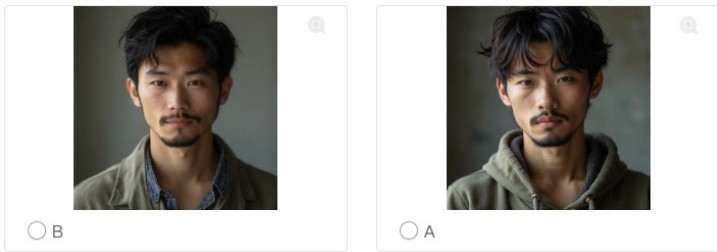

Figure 19: **An example question from the user study.** Given a pair of source and target images, along with an original image and two edited images, participants were asked to select the image that demonstrated superior editing quality while preserving the original identity.

