# OpenReview forum: "PairEdit: Learning Semantic Variations for Exemplar-based Image Editing"
_NeurIPS.cc/2025/Conference — NeurIPS 2025 poster_

### Official Review · Reviewer_1DPG · 2025-06-30

**Clarity:** 2
**Significance:** 3
**Originality:** 3
**Rating:** 4
**Confidence:** 3

**Summary:**

This paper propose PairEdit, a novel exemplar-based image editing framework that learns semantic transformations directly from a small number of source-target image pairs without relying on any textual descriptions. The method introduces a dual-LoRA architecture, consisting of a content LoRA that reconstructs the source image and a semantic LoRA that captures the semantic variation between the paired images. By modeling the difference in noise predictions as a guidance direction, and introducing a content-preserving noise schedule, PairEdit enables stable and effective disentanglement of content and semantics. During inference, only the semantic LoRA is applied to the diffusion model, allowing for precise and continuous editing from the same noise seed while preserving content structure. The framework supports high-quality editing, continuous control via LoRA scaling, and composable edits through LoRA merging. Extensive qualitative, quantitative, and user study evaluations demonstrate that PairEdit outperforms existing exemplar-based methods in both semantic fidelity and identity preservation, despite using minimal training data.

**Questions:**

1. Since each semantic concept requires training a separate LoRA, could the authors comment on whether they envision a way to share or transfer knowledge across related edits (e.g., glasses and sunglasses)? Is it possible to extend the current framework to support multi-attribute or compositional learning without retraining separate LoRAs?

2. The method relies on manually curated or semi-automatically generated image pairs. Could the authors clarify how scalable this pairing process is in practice? Would the model still work effectively with noisy or weakly aligned pairs, or is high-quality alignment essential?

3. Continuous editing via LoRA scaling is one of the key features of PairEdit, a more thorough ablation or sensitivity analysis of scaling values across semantics would significantly strengthen the claim. Have the authors evaluated how scaling affects visual quality and identity preservation across a wider range of γ values?

**Ethical Concerns:**

["NO or VERY MINOR ethics concerns only"]

**Final Justification:**

The rebuttal satisfactorily addressed the key issues I raised. Although some fundamental limitation exists due to the pipeline of the method, the paper makes a valuable contribution. I’m keeping my original score.

**Limitations:**

Yes

**Paper Formatting Concerns:**

I didn't find any major formatting issues.

**Quality:**

3

**Strengths And Weaknesses:**

Strengths:
1. The paper introduces a fully visual approach to exemplar-based editing, eliminating the need for textual prompts or language models. This is a significant advancement in editing controllability, especially in scenarios where language fails to capture nuanced visual changes. The ability to learn from 1–3 image pairs demonstrates high data efficiency and reduces reliance on large annotated datasets.

2. The dual-LoRA architecture, jointly training content and semantic LoRAs on the same base model, offers a clean and modular separation of identity-preserving reconstruction and semantic transformation. This not only improves visual fidelity but also enhances interpretability and edit-specific reusability, which is both technically sound and practically beneficial.

3. The proposed formulation of semantic variation as a guidance direction in noise space, inspired by classifier-free guidance, is both mathematically grounded and intuitively appealing. The introduction of a content-preserving noise schedule to stabilize this process reflects thoughtful model design and facilitates reliable training under limited supervision.

Weaknesses

1. Each semantic variation requires its own LoRA to be trained from scratch. There is no shared representation or meta-learning mechanism to transfer knowledge across similar edits. This limits the scalability of the approach in multi-attribute editing scenarios and makes it less practical for real-world systems with a large or open-ended set of editing intents.

2. The training image pairs are either manually curated or generated through semi-automatic pipelines involving region transfer and SDEdit. This limits the automation and reproducibility of the method, and introduces a potential bottleneck when scaling to large or diverse datasets. Additionally, pair alignment quality directly affects performance, introducing variability.

3. While PairEdit supports continuous control of semantic strength via LoRA scaling, the method lacks a thorough analysis of how sensitive the editing behavior is to the choice of scaling factors. There is no principled guideline for setting this parameter, and over- or under-scaling may lead to either negligible or distorted edits. This raises concerns about consistency, interpretability, and usability across different semantics.

---

> ### Author Rebuttal · Authors · 2025-07-31
>
> We thank Reviewer 1DPG for the insightful feedback! We are encouraged that the reviewer finds our method to be both mathematically grounded and intuitively appealing. We are pleased that the reviewer recognizes the strength of eliminating textual prompts and the high data efficiency of our method.
>
> Below, we address the reviewer’s concerns point by point.
>
> > **W1 & Q1: Each semantic variation requires its own LoRA to be trained from scratch. Is it possible to transfer knowledge across similar edits? Is it possible to support multi-attribute learning without retraining separate LoRAs?**
>
> Thank you for the suggestion. We tested transferring a learned LoRA (e.g., glasses) to another edit (e.g., sunglasses or even eye size). It successfully learns the new edits and reduces optimization steps from 500 to 50. This demonstrates that the knowledge of the learned LoRAs can be effectively reused to accelerate the learning of related edits, showing strong transferability and generalization capabilities. We appreciate the reviewer’s valuable suggestion and will definitely include these results. As we cannot upload images in the rebuttal, we will include these new results in the revised paper.
>
> Regarding multi-attribute learning, one approach is to use image pairs with multi-attribute variations, allowing our model to simultaneously learn these variations. Another approach is to apply the suggested transfer learning method to accelerate the multi-attribute learning process. Additionally, as described in L242 and Figure 6, our method supports multi-attribute edits by merging the learned LoRAs.
>
>
> > **W2 & Q2: The training image pairs are manually curated or generated through semi-automatic pipelines. This limits the automation and reproducibility of the method. Could the authors clarify how scalable this pairing process is in practice? Would the model work effectively with noisy or weakly aligned pairs, or is high-quality alignment essential?**
>
> We acknowledge that the current training image pairs rely on semi-automatic pipelines. However, we would like to clarify that generating training image pairs is actually quite convenient in practice. We simply copy the target region coarsely into the source image. Even though the pasted target images often have obvious artifacts around the boundaries, we find that the model is able to effectively learn the semantic edits while ignoring these inconsistencies.
>
> For weakly aligned pairs, our model can still learn semantic variations. However, it may learn additional unintended semantics, as it does not explicitly know which semantics are targeted given that only 1–3 pairs are used for training. High-quality image pairs result in improved quality of generations, particularly regarding identity preservation. As we cannot upload images in the rebuttal, we will include results using weakly aligned pairs in the revised paper.
>
> Moreover, although our method relies on paired images, we would like to clarify that this task setting is meaningful and applicable in real-world scenarios where users have a clear and specific visual goal but cannot accurately describe it using text. For example, users may want to modify specific facial attributes (e.g., nose shape or ear shape) to match a very specific target look. These edits are often difficult to describe precisely with text but can be easily conveyed through exemplar image pairs. In fact, the second row of Figure 3 in our paper demonstrates a real-world case from a GitHub issue of Slider [1], submitted by a user seeking assistance with exemplar image pairs—a case that could not be solved by Slider but is successfully handled by our method.
>
>
> > **W3 & Q3: While PairEdit supports continuous control of semantic strength via LoRA scaling, the paper lacks a thorough analysis of how sensitive the editing behavior is to the choice of LoRA scaling factors. There is no principled guideline for setting this parameter, and over- or under-scaling may lead to either negligible or distorted edits. This raises concerns about consistency, interpretability, and usability across different semantics. Have the authors evaluated how scaling affects visual quality and identity preservation across a wider range of γ values?**
>
> Thank you for pointing this out. Indeed, over- or under-scaling factors may result in negligible or distorted edits. In particular, under-scaling factors lead to negligible editing effect, while over-scaling factors may introduce artifacts or degrade image quality. According to our observations, setting the LoRA scaling factor between 0.3 and 1.0 consistently yields high-quality editing results with varying levels of editing strength. For different types of semantic editing, we typically set the scaling factor to 0.6, which empirically achieves an optimal balance between editing quality and identity preservation.
>
> **References**
>
> [1] Gandikota et. al. "Concept Sliders: LoRA Adaptors for Precise Control in Diffusion Models". ECCV, 2024.

---

> > ### Comment · Reviewer_1DPG · 2025-08-05
> >
> > Thank you for the detailed and constructive rebuttal. I appreciate that most of my concerns were addressed. I’m especially looking forward to seeing the visual outputs for question 1 in the revised version.

---

> > > ### Author Response · Authors · 2025-08-05
> > > **Response to Reviewer 1DPG**
> > >
> > > Dear Reviewer 1DPG,
> > >
> > > We are happy to hear that our rebuttal has addressed your concerns well! We sincerely appreciate your support for our work, especially your valuable suggestion regarding transferring a learned LoRA to another edit. This approach has notably reduced our training time and enhanced the practical flexibility and efficiency of our method. We will definitely include this new experiment in the revised paper.
> > >
> > > If you have any further questions or suggestions, please do not hesitate to let us know.
> > >
> > > Best regards,
> > >
> > > The Authors

---

### Official Review · Reviewer_ahKQ · 2025-07-03

**Clarity:** 3
**Significance:** 3
**Originality:** 3
**Rating:** 4
**Confidence:** 3

**Summary:**

This paper introduces PairEdit, a novel visual editing method that learns complex editing semantics from a limited number of image pairs (even a single pair) without relying on textual guidance.

**Questions:**

1. I am wondering if exemplar-based image editing has some real-world application scenarios?
2. I am also wondering the effect of proposed method on other backbones.

**Ethical Concerns:**

["NO or VERY MINOR ethics concerns only"]

**Final Justification:**

The rebuttal adequately addressed the main concerns.

Reliance on paired images: The authors clarified that creating such pairs is feasible in practice and provided examples where text-based editing would fall short. While the method still cannot operate in unpaired settings, the clarification helps justify the relevance of their task setup.

Limited backbones tested: The authors added results using SDXL, which demonstrates some degree of generalization. Although the performance is not as strong as with FLUX, the explanation is reasonable and the new results will be included in the revision.

Real-world applicability: The authors provided concrete use cases (e.g., fine-grained facial edits), which support the practical motivation of exemplar-based editing.

That said, the scope of the method is still somewhat narrow, and the generalization beyond the studied setting remains limited. Given that, I am maintaining my score at 4.

**Limitations:**

The proposed method requires an image pair to complete the editing process, which may limit its generalization ability.

**Quality:**

3

**Strengths And Weaknesses:**

Strength
1. No Textual Guidance Needed: A key strength is that PairEdit eliminates the need for text prompts to describe editing semantics, which can be challenging to specify precisely using text alone. This makes it more practical for certain editing tasks.
2. Effective Learning from Limited Data: The method can effectively learn intricate semantics from a very small dataset, even a single image pair. This is a significant advantage over methods that require large datasets or extensive manual annotation.
3. Extensive experiments and good results: The paper has extensive quantitative and qualitative experiments and the results shows the effectiveness of the method with high fidelity.

Weakness
1. Reliance on Paired Images: The most significant limitation explicitly stated by the authors is its "reliance on paired images, which is not directly applicable to unpaired datasets". This limits its applicability to scenarios where such pairs can be readily obtained. Especially for real-world images captured by phones.
2. Limited backbones tested: The method is tested only with the FLUX.1-dev and LoRA adapters; its effectiveness on other diffusion models or fine-tuning schemes remains untested.

---

> ### Author Rebuttal · Authors · 2025-07-31
>
> We thank Reviewer ahKQ for the insightful feedback! We are encouraged that the reviewer finds our results to be good, and recognizes the strength of eliminating textual guidance and effective learning from limited data.
>
> Below, we address the reviewer’s concerns point by point.
>
> > **W1: Reliance on Paired Images: The most significant limitation explicitly stated by the authors is its "reliance on paired images, which is not directly applicable to unpaired datasets". This limits its applicability to scenarios where such pairs can be readily obtained.**
>
> Indeed, we stated in the limitations section that our method relies on paired images. However, we would like to clarify that generating training image pairs is actually quite convenient in practice. We simply copy the target region coarsely into the source image. Even though the pasted target image often exhibits obvious artifacts around the boundary, we find that the model effectively learns the semantic edits while ignoring these artifacts.
>
> Moreover, although our method relies on paired images, this task setting is meaningful and applicable to real-world scenarios. For example, users may want to change the nose shape to a very specific target shape that cannot be precisely described using text alone. Our method is well-suited for such fine-grained, visually guided edits. In fact, the second row of Figure 3 (i.e., the example of changing the nose shape) illustrates a real case taken from a GitHub issue for Slider [1], submitted by a user seeking assistance with exemplar image pairs.
>
> Note that existing exemplar-based editing methods, such as Slider [1] and Edit Transfer [2], also rely on paired images. To the best of our knowledge, there are currently no methods capable of performing attribute-specific visual edits in an unpaired setting. Developing models capable of learning from unpaired data remains an open challenge in this domain, and it is one of the key directions we intend to pursue in future work.
>
> > **W2 & Q2: Limited backbones tested: The method is tested only with the FLUX.1-dev and LoRA adapters; its effectiveness on other diffusion models or fine-tuning schemes remains untested.**
>
> We appreciate the reviewer’s suggestion. We built our model upon FLUX.1-dev because it is one of the current state-of-the-art open-source image generation models. Following the reviewer’s recommendation, we additionally tested our method with SDXL as the backbone. We found that the model remains capable of learning meaningful editing semantics. However, its performance is not as strong as with FLUX in terms of identity preservation and editing quality. We hypothesize that this performance gap stems from differences in the mathematical modeling of the backbone models—our approach is theoretically grounded in the noise formulation (Eq. 1) used in FLUX from a flow-matching perspective, whereas SDXL employs a DDPM noise schedule. As we cannot upload images in the rebuttal, we will include the results using SDXL in the revised version of the paper. Regarding the use of LoRA adapters, we adopted them due to their lightweight nature and fine-tuning efficiency. LoRA allows us to adapt the backbone with a relatively small number of trainable parameters, which is especially important when training on limited paired data. It also helps maintain the general generation capabilities of the pretrained model and is compatible with other learned LoRAs available in the community.
>
>
> > **Q1: What are the real-world application scenarios for exemplar-based image editing?**
>
> Exemplar-based image editing is particularly valuable in real-world scenarios where users have a clear and specific visual goal but cannot accurately describe it using text. For example, users may want to modify specific facial attributes (e.g., nose shape or ear shape) to match a very specific target look. These edits are often difficult to describe precisely with text but can be easily conveyed through exemplar image pairs. In fact, the second row of Figure 3 in our paper demonstrates a real-world case from a GitHub issue of Slider [1], submitted by a user seeking assistance with exemplar image pairs—a case that could not be solved by Slider but is successfully handled by our method. In addition to facial editing, exemplar-based image editing may also apply to design-related scenarios, such as character and product design, where designers frequently need to adjust visual appearances to match very specific target looks.
>
> > **Q2: The effect of the proposed method on other backbones.**
>
> Please refer to our detailed response to "W2 & Q2".
>
>
>
> **References**
>
> [1] Gandikota et. al. "Concept Sliders: LoRA Adaptors for Precise Control in Diffusion Models". ECCV, 2024.
>
> [2] Chen et. al. "Edit Transfer: Learning Image Editing via Vision In-Context Relations". arXiv:2503.13327, 2025.

---

> > ### Comment · Reviewer_ahKQ · 2025-08-07
> >
> > The authors responded clearly to the main concerns.
> >
> > They clarified the process of generating paired images and gave a practical example where such exemplar edits are needed but difficult to describe using text. This helps justify the relevance of their setup, though the method still cannot generalize to unpaired settings.
> >
> > Additional results with SDXL were provided, showing the method can be applied beyond FLUX, though with reduced performance. The explanation for this difference is plausible, and including these results in the final paper will help.
> >
> > The real-world applications of exemplar-based editing were explained with specific use cases, which adds to the clarity of the paper.
> >
> > Overall, while the method is technically sound and the authors addressed the concerns responsibly, the limited scope of applicability remains. I am keeping my score at 4.

---

### Official Review · Reviewer_aUji · 2025-07-03

**Clarity:** 3
**Significance:** 3
**Originality:** 3
**Rating:** 4
**Confidence:** 3

**Summary:**

This paper introduces PairEdit, a novel visual editing method that learns complex editing semantics from paired source-target image examples without relying on textual descriptions. Unlike existing exemplar-based methods that depend on text prompts, PairEdit utilizes target noise prediction to explicitly model semantic changes between paired images through a guidance direction term. It also employs a content-preserving noise schedule to enhance semantic learning, and optimizes distinct LoRAs to separate semantic variation learning from content retention. Extensive evaluations show that PairEdit effectively captures intricate semantics while greatly improving content consistency compared to existing methods.

**Questions:**

See weaknesses.

**Ethical Concerns:**

["NO or VERY MINOR ethics concerns only"]

**Final Justification:**

Thanks for the rebuttal. I keep my score to accept this paper.

**Limitations:**

yes

**Quality:**

3

**Strengths And Weaknesses:**

Strengths:
- The proposed method is remarkably efficient, requiring only a few image pairs or even a single pair for training. This efficiency significantly reduces the amount of data necessary for effective model training, which is a notable advantage in resource-limited scenarios.
- It delivers strong performance results, with the generated images being particularly impressive in terms of quality and fine-grained detail. This highlights the method's capability to produce high-quality edits that are visually striking.
- The approach is both innovative and well-founded, addressing a clear gap in the field with a novel solution. The methodology is logically structured and motivated, making it a significant contribution to the area of image editing.

Weaknesses:
- To provide a more comprehensive evaluation, it would be beneficial to include examples of less successful cases. This would help illustrate the method's limitations and identify potential areas for further improvement or refinement.
- The evaluation of the proposed method is currently limited to the FLUX.1-dev model. Testing on additional models would help verify the general applicability and robustness of the method across different scenarios and datasets.
- The quantitative comparisons are primarily based on LPIPS and CLIP distance metrics. Expanding the range of evaluation metrics could provide a more thorough assessment of the method's performance, capturing various aspects of image quality and semantic coherence beyond these two metrics.

---

> ### Author Rebuttal · Authors · 2025-07-31
>
> We thank Reviewer aUji for the insightful feedback! We are encouraged that the reviewer finds our method innovative, well-founded, and efficient. We are pleased the reviewer recognizes the strong performance of our model.
>
> Below, we address the reviewer’s concerns point by point.
>
> > **W1: To provide a more comprehensive evaluation, it would be beneficial to include examples of less successful cases.**
>
> Thank you for this helpful suggestion. We agree that including failure cases is important for a more comprehensive and transparent evaluation. Indeed, our method does have less successful cases, particularly when the source and target images involve dramatic structural changes. For instance, edits involving a person transitioning from standing to jumping, or transforming a cat into a dog, often exceed the intended scope of our method. Due to rebuttal constraints, we are unable to upload visual examples of less successful cases here. However, we will add a dedicated section in the revised paper to analyze and illustrate these less successful cases.
>
>
> > **W2: The proposed method is evaluated using the FLUX.1-dev model. Testing on additional models would help verify the general applicability and robustness of the method across different scenarios and datasets.**
>
> We appreciate the reviewer’s suggestion. We built our model upon FLUX.1-dev because it is one of the current state-of-the-art open-source image generation models. Following the reviewer’s recommendation, we additionally tested our method with SDXL as the backbone. We found that the model remains capable of learning meaningful editing semantics. However, its performance is not as strong as with FLUX in terms of identity preservation and editing quality. We hypothesize that this performance gap stems from differences in the mathematical modeling of the backbone models—our approach is theoretically grounded in the noise formulation (Eq. 1) used in FLUX from a flow-matching perspective, whereas SDXL employs a DDPM noise schedule. As we cannot upload images in the rebuttal, we will include the results using SDXL in the revised version of the paper.
>
> > **W3: The quantitative comparisons are primarily based on LPIPS and CLIP distance metrics. Expanding the range of evaluation metrics could provide a more thorough assessment of the method's performance.**
>
> Thank you for this valuable suggestion. We agree that using multiple metrics provides a more thorough evaluation. In the current version of the paper, we followed Slider [1] by employing LPIPS and CLIP metrics for quantitative evaluation. Following the reviewer’s advice, we have conducted additional quantitative experiments using the DINO metric [2], which has been shown to effectively capture semantic similarity and structural consistency in vision tasks. The results are presented in the table below. As demonstrated, our method consistently outperforms the baselines under the DINO metric, further validating the effectiveness of our approach. We thank the reviewer for this suggestion and will definitely include these new results in the revised paper.
>
> | DINO Score ↑  | SDEdit  | Textual Slider | Visual Slider | Ours |
> | -| - | -| -|-|
> | Chubbiness |   0.8420   |  0.9152   |  0.8882   |  **0.9588**   |
> | Glasses |  0.8408    |  0.8951 | 0.8810  |  **0.9065** |
> | Smile |  0.9086    |  0.8853  	|  0.9143 |  **0.915** |
> | Age |  0.8597  |  0.8875  |  0.8564 |  **0.8885** |
>
> **References**
>
> [1] Gandikota et. al. "Concept Sliders: LoRA Adaptors for Precise Control in Diffusion Models". ECCV, 2024.
>
> [2] Caron et. al. "Emerging properties in self-supervised vision transformers". ICCV, 2021.

---

### Official Review · Reviewer_gUZ4 · 2025-07-07

**Clarity:** 3
**Significance:** 2
**Originality:** 2
**Rating:** 4
**Confidence:** 4

**Summary:**

This paper tackles the problem of exemplar-based image editing. It takes in several images pairs of the same editing as the exemplar and will output the same editing given new images. It is in a test-time finetuning manner with two sets of learnable parameters:  Content LoRA for identity keeping and semantic LoRA for semantic of the edits. It proposes a CFG suitable for the two sets of LoRA and reformulates the CFG formula customized for the rectified flow setting. Also, it reset the temporal scheduling for content preservation purposes.

**Questions:**

1. I think the most fair comparison is to compare with Pair Customization with the same backbone and same num of training example (suggesting 1 training pair for practice)

2. If the editing task given complex context should be moved to MLLM structure for better study since they have longer context window and better contexts understanding capability?

**Ethical Concerns:**

["NO or VERY MINOR ethics concerns only"]

**Final Justification:**

addressed my concern.

**Limitations:**

See weakness.

**Quality:**

2

**Strengths And Weaknesses:**

Strengths:
1. The paper is easy to follow and well written.

2. The algorithm looks promising because the performance seems better than previous methods. (though unclear to me)

3. The authors have mentioned the limitation.

Weakness

1.	Comparison unfair. The comparison methods use SD-1.5 backbone, which is weaker than Flux1-dev that this paper uses, and VISII & Analogist are designed for zero training, which has different speed and computes requirement than this paper. Therefore, those comparison methods do not provide enough information for me to judge the significance of this paper. I think the closest comparison method is Pair Customization, which also use Two LoRAs (content + style) as this paper did. I think the most fair comparison is to compare with Pair Customization with the same backbone and same num of training example.

2.	The idea wise is lacking novelty. LoRA for content and semantic is similar to Pair Customization, and the temporal scheduling seems common in the diffusion-based image editing model.

3.	The task setting is less common. Given three pairs of the images with the same edit. It is hard to find the same edits of more than one pairs. Do the glass of the three exemplar pairs in Fig. 1 need to be the same look? If yes, it is hard to get those editing pairs in my view. So the practical use is only for one exemplar pair, which is less effective.

---

> ### Author Rebuttal · Authors · 2025-07-31
>
> We thank Reviewer gUZ4 for the insightful feedback! We are encouraged that the reviewer finds our method promising and our paper well-written.
>
> Below, we address the reviewer’s concerns point by point.
>
> > **W1 & Q1: The comparison methods use the SD-1.5 backbone, which is weaker than FLUX.1-dev used in this paper. VISII & Analogist are designed for zero training, which has different speed and computation requirements compared to this paper. I think the fairest comparison is to compare with Pair Customization using the same backbone and the same number of training examples (e.g., one training pair for practice).**
>
> Thank you for the suggestion. We address this concern through the following points:
>
> **1. On the fairness of using FLUX.1-dev for baselines:**
>
> We would like to clarify that two of the main baselines in our paper—Slider [1] and Edit Transfer [2]—are both implemented using the same FLUX.1-dev backbone as our method. Note that the comparison with Edit Transfer [2] is presented in the Appendix (Figure 8).
>
> **2. On including VISII and Analogist as baselines:**
>
> Due to the limited number of exemplar-based image editing methods with open-source implementations, we included VISII and Analogist for broader context. For VISII, we would like to clarify that it is not a zero-training method. It optimizes semantic edits, represented by the source and target images, into the textual instruction space of InstructPix2Pix. The VISII paper states that it requires training for 1,000 steps per target semantic. For reference, our method takes approximately 8 minutes for optimization, while VISII takes approximately 10 minutes. Regarding Analogist, we acknowledge that it is a training-free method. Following the reviewer’s suggestion, we will replace Analogist in Figure 3 with another optimization-based method, Edit Transfer [2]. Note that the comparison with Edit Transfer is already included in the Appendix (Figure 8).
>
> **3. On adding Pair Customization as a baseline:**
>
> Thank you for suggesting Pair Customization as a baseline. Following your recommendation, we compared our method with Pair Customization using the same FLUX.1-dev backbone and trained both methods with only one image pair. For simple edits (e.g., "smile"), Pair Customization can learn the editing semantics but significantly alters the identity of the source image. For complex edits (e.g., nose shape change in the second row of Figure 3), Pair Customization fails to learn the editing semantics. It is also important to note that Pair Customization requires a text prompt as auxiliary input, whereas our method is fully prompt-free, relying only on visual exemplars. As we cannot upload visual results in the rebuttal, we present quantitative results below, showing that our method achieves significantly lower LPIPS distances compared to Pair Customization. We will include the visual comparisons in the revised version of the paper.
>
> Moreover, the insights of our method and Pair Customization are fundamentally different. Pair Customization adopts a content/style LoRA architecture to learn the style of the reference image, while our method introduces a guidance-based semantic variation loss to directly learn the semantic variation between the source and target images. For a more detailed discussion of the differences, please refer to our response to W2 below.
>
> |   | LPIPS (PairCustom) ↓ |   LPIPS (Ours, single pair) ↓ | CLIP (PairCustom) ↑ |CLIP (Ours, single pair) ↑ |
> | -| - |-| - |-|
> |  Smile 			|  0.1964 | 0.1014  |  0.2542 |  0.2849 |
> | Chubbiness 	| 0.2270   |0.0767   | 0.2281  | 0.2365  |
> | Glasses   	  	| 0.1597  | 0.1076  | 0.2367  |	0.2826 |
> |  Age       		| 0.1154   | 0.0828  | 0.2199  |0.2234|
>
> > **W2: LoRA for content and semantic is similar to PairCustom. The temporal scheduling seems common in diffusion-based image editing models.**
>
> We would like to clarify the misunderstanding regarding the insights and key contributions of our work. The content/semantic LoRA module and temporal noise scheduling strategy are NOT the key contributions of our work. Our ablation study (Figure 7) demonstrates that removing either the content/semantic LoRA module or the proposed noise scheduling (Eq. 8) still enables the model to learn semantic edits effectively. In fact, the content/semantic LoRA module is used to enhance content-semantic disentanglement, while the noise scheduling improves generalization capability. Additionally, we have acknowledged (Line 122 of our paper) that the design of the content/semantic LoRA module is inspired by Pair Customization.
>
> The primary contribution of our work is the guidance-based semantic variation loss, formally derived from Eq. (4) to Eq. (10). This loss explicitly models the semantic variation between source and target images using a guidance-based direction term. As shown in our ablation study (Figure 7), this loss, rather than the content/semantic LoRA module, enables effective learning of the editing semantics exampled by the paired images. To the best of our knowledge, our method is the first to accurately learn complex and fine-grained semantic edits from a single exemplar image pair.
>
>
> > **W3: Given three image pairs with the same edit, it's difficult to find the same edits for more than one pair. Do the glasses in the three exemplar pairs in Fig. 1 need to be identical? If yes, it is hard to get those image pairs in my view. So the practical use is only for one exemplar pair, which is less effective.**
>
> Thank you for raising this point. We would like to clarify that the glasses in the three exemplar pairs shown in Figure 1 do not need to be identical; indeed, we used different glasses for this example. Zooming in on Figure 1 shows variations in shape of the glasses. We will include all training pairs in the revised paper.
>
> Regarding the number of training pairs, our method can learn meaningful edits from a single exemplar pair for all semantic edits presented in the paper. Section E in the Appendix (Figure 11) presents comparisons using only one image pair. However, multiple pairs improve our model's generalization capability. For instance, in the “glasses” example, using three image pairs allows generalization of the “add glasses” edit even to a dog image, difficult with only one pair.
>
> We appreciate the reviewer’s suggestion, and will highlight these clarifications explicitly in the revised paper.
>
> > **Q1: I think the fairest comparison is to compare with Pair Customization with the same backbone and same num of training example (suggesting 1 training pair for practice)**
>
> Please refer to our detailed response to "W1 & Q1" above.
>
> > **Q2: For editing tasks involving complex context, should the method be moved to an MLLM structure for better study since they have longer context window and better contexts understanding capability?**
>
> Thank you for this insightful question. We agree that MLLM-based image editing methods excel in tasks needing complex contextual understanding. However, our work specifically targets exemplar-based image editing with FLUX, defining edits visually through before-and-after image pairs rather than textual prompts or multimodal context. This setting is complementary to the task of MLLM-based editing and is particularly effective when visual editing goals (e.g., very specific fine-grained regional changes) are difficult to describe using text. We believe both tasks are valuable and complementary in practice. Exploring integration with MLLMs is a promising direction we plan for future work.
>
> Additionally, Figure 8 in the Appendix compares our method with GPT-4o, which supports MLLM-based image editing, showing that our method achieves superior identity preservation and editing quality.
>
> **References**
>
> [1] Gandikota et. al. "Concept Sliders: LoRA Adaptors for Precise Control in Diffusion Models". ECCV, 2024.
>
> [2] Chen et. al. "Edit Transfer: Learning Image Editing via Vision In-Context Relations". arXiv:2503.13327, 2025.

---

> > ### Comment · Reviewer_gUZ4 · 2025-08-04
> > **Reply to the rebuttal**
> >
> > Thanks for the rebuttal. It addressed my concerns and will raise my score.

---

> ### Author Response · Authors · 2025-08-04
> **Response to Reviewer gUZ4**
>
> Dear Reviewer gUZ4,
>
>
> We are happy to hear that our rebuttal has addressed your concerns well! We sincerely appreciate your support for our work. If you have any further questions or suggestions, please do not hesitate to let us know.
>
>
> Best regards,
>
> The Authors

---

### Decision · Program_Chairs · 2025-09-17

**Decision:**

Accept (poster)

**Comment:**

This paper addresses exemplar-based image editing: it takes multiple image pairs demonstrating the same edit as exemplars and applies that edit to new images. Using test-time finetuning, it employs two learnable parameter sets—Content LoRA (to preserve image identity) and Semantic LoRA (to capture edit semantics). The method introduces a Classifier-Free Guidance (CFG) strategy tailored to these LoRAs, reformulates the CFG formula for rectified flow settings, and resets temporal scheduling to enhance content preservation.

All the reviewers hold positive reviews and most concerns are solved. The authors need to add the rebuttal experiments into the final draft.